# Povidone-Iodine and Hydrogen Peroxide Combination Improves the Anti-Biofilm Activity of the Individual Agents on *Staphylococcus aureus*

**DOI:** 10.3390/ijms26094390

**Published:** 2025-05-06

**Authors:** Le Wan, Jaishree Sankaranarayanan, Chan-Young Lee, Hongyan Zhou, Taek-Rim Yoon, Jong-Keun Seon, Kyung-Soon Park

**Affiliations:** 1Department of Orthopedic Surgery, Center for Joint Disease, Chonnam National University Medical School and Hospital, Hwasun-gun 58128, Republic of Korea; wle202302@gmail.com (L.W.); jaanu.p2206@gmail.com (J.S.); cnuhoslee@gmail.com (C.-Y.L.); imccnuh@gmail.com (T.-R.Y.); seonbell@jnu.ac.kr (J.-K.S.); 2Department of Heart Research Center, Chonnam National University Medical School and Hospital, Gwangju 61469, Republic of Korea; zhouhy202111@gmail.com

**Keywords:** *Staphylococcus aureus*, povidone-iodine (PVP-I), hydrogen peroxide (H_2_O_2_), biofilms, anti-biofilm activity

## Abstract

*Staphylococcus aureus*, particularly methicillin-resistant *S. aureus* (MRSA), poses significant challenges in healthcare settings due to its ability to form biofilms on various surfaces. These biofilms enhance bacterial survival and increase resistance to conventional treatments, complicating infection control efforts. This study evaluated the efficacy of combined povidone-iodine (PVP-I) and hydrogen peroxide (H_2_O_2_) to disrupt pre-formed *S. aureus* biofilms. A series of assays—including crystal violet staining, colony-forming unit (CFU) enumeration, gene expression analysis, and confocal laser scanning microscopy—were performed to assess the effects of each treatment individually and in combination. The combined treatment resulted in significantly greater reductions in biofilm biomass and viable bacteria compared with either agent alone. Gene expression analysis revealed downregulation of key biofilm-associated genes (*icaA*, *icaB*, *icaD*, *icaR*, and *clfA*), suggesting interference with biofilm stability and maintenance. While formal synergy quantification was not conducted, the observed effects suggest a potentially synergistic or additive interaction between the two agents. These findings support the use of dual antiseptic strategies as a promising approach to biofilm eradication and highlight the potential clinical utility of dual antiseptic strategies. However, we underscore the need for further optimization and safety evaluation.

## 1. Introduction

Periprosthetic joint infection (PJI) is one of the most severe complications following artificial joint replacement (AJR) [1]. Bacterial pathogens are responsible for over 98% of PJIs, with *S. aureus* and coagulase-negative staphylococci accounting for 50–60% of all cases [2,3]. *S. aureus* is the leading cause of infections related to implanted biomaterials [4]. In particular, methicillin-resistant *S. aureus* continues to pose a significant global health challenge in hospital settings. This Gram-positive bacterium is a leading cause of hospital-associated infections (HAIs), substantially contributing to patient morbidity and mortality and escalating healthcare costs worldwide.

The prevalence of MRSA considerably varies across different countries and regions. According to the 2014 World Health Organization (WHO) report on antimicrobial resistance, the prevalence of MRSA infections in hospitals exceeded 20% in all regions, with some areas reporting infection rates as high as 80% [5]. In South Korea, the burden of hospital-acquired *S. aureus* bloodstream infections (SA-BSIs) is notably high. As per a national study conducted in 2011, the estimated incidence of hospital-acquired MRSA-BSIs was 0.12 cases per 1000 patient days, while that of methicillin-sensitive SA-BSIs (MSSA-BSIs) was 0.04 cases per 1000 patient days. This translates to approximately 2946 MRSA-BSI cases and 986 MSSA-BSI cases annually in hospitals with more than 500 beds. The mortality rate associated with these infections is also alarmingly high (31.9% for MRSA-BSI cases and 14.1% for MSSA-BSI cases). Moreover, the economic impact of these infections is significant, with the additional cost per case estimated to be USD 20,494 for MRSA-BSIs and USD 6914 for MSSA-BSIs, leading to a total cost of more than USD 67.2 million annually. In recent years, the prevalence of MRSA in South Korean hospitals has increased, with MRSA accounting for 60–70% of all *S. aureus* infections. This high prevalence underscores the urgent need for effective infection control measures within healthcare facilities to mitigate the spread and impact of MRSA [6].

Recent studies have highlighted the potential of PVP-I and H_2_O_2_ as effective agents in the prevention and control of bacterial and fungal biofilm-associated infections in clinical settings. The use of PVP-I and H_2_O_2_ has shown considerable potential in addressing these infections. The use of PVP-I and H_2_O_2_ offers several advantages over traditional antimicrobial methods, particularly in managing biofilm-associated infections. Iodine-based compounds, such as PVP-I, have long been recognized for their broad-spectrum antimicrobial activity. They are effective against a wide range of pathogens, including antibiotic-resistant strains (e.g., MRSA). These agents act by releasing free iodine, which penetrates microbial cell walls and disrupts critical cellular processes, resulting in bacterial death. Their efficacy in reducing biofilm formation makes them particularly valuable in hospital settings [7,8].

Biofilms are complex systems consisting of sessile bacteria encased within an extracellular polymeric substance (EPS). EPS serves as the structural scaffold of biofilms. It also acts as a physical and chemical barrier and makes it difficult for antibiotics to penetrate biofilms, thereby forming a protective shield against antimicrobial agents [9,10]. H_2_O_2_ is a potent oxidizing agent that generates free radicals, causing oxidative stress and damage to bacterial cells. It is effective against various hospital pathogens, including *S. aureus*. Given its ability to penetrate biofilms and its relatively low toxicity to human cells at appropriate concentrations, H_2_O_2_ is an attractive option for infection control [11].

To address the persistent challenge of biofilm-associated infections, this study investigated the combined application of PVP-I and H_2_O_2_—two clinically available antiseptics with distinct mechanisms of action—for enhanced eradication of mature *S. aureus* biofilms [12,13,14]. This combined approach can overcome the limitations of monotherapy, offering a more comprehensive solution to combat *S. aureus* infections [15]. By targeting different aspects of bacterial survival and proliferation, this strategy may enhance overall efficacy and reduce the risk of developing resistance. Moreover, both agents used in this method are cost-effective and readily available, making them practical for implementation in hospital infection control programs. Their use could significantly reduce infection rates, thereby lowering patient morbidity and mortality, shortening hospital stays, and alleviating the economic burden on healthcare systems.

This study assessed both individual and combined effects of PVP-I and H_2_O_2_ on the eradication of pre-formed *S. aureus* biofilms. Furthermore, we evaluated changes in the expression of key biofilm-associated genes to elucidate potential molecular mechanisms of disruption. By leveraging the complementary actions of these agents, we aimed to develop a more effective strategy against biofilm-related infections, contributing to improved infection control in hospital settings.

## 2. Results

### 2.1. Confirmation of S. aureus Identity Using Morphological and Culture-Based Characteristics

*S. aureus* was analyzed using mannitol salt agar (MSA) and by performing Gram staining. On microscopic examination, the Gram-positive bacterium *S. aureus* appeared purple and exhibited a grape-like cluster arrangement, which is crucial for distinguishing it from other bacterial species. MSA is both selective and differential; it is used to isolate and identify *S. aureus*. It contains mannitol, which can be fermented by *S. aureus*. The production of acid during the fermentation process lowers the pH and changes the color of the phenol red indicator, resulting in yellow colonies. This color change differentiates *S. aureus* from other non-mannitol-fermenting staphylococci, such as coagulase-negative staphylococci, which form pink colonies on MSA. These morphological and biochemical characteristics are critical for diagnosing and managing infections in clinical microbiology (Figure 1).

In addition to morphological and culture-based identification, the biofilm-forming ability of the strain was preliminarily assessed using crystal violet staining after 48 h of incubation. The absorbance at 570 nm indicated significant biofilm biomass compared to the negative control, confirming that the strain is a strong biofilm producer suitable for subsequent antibiofilm assays (Figure 2A).

### 2.2. Quantitative Evaluation of Biofilm Disruption by PVP-I and H_2_O_2_

The effectiveness of PVP-I and H_2_O_2_, both individually and in combination, in disrupting pre-formed biofilms was assessed using crystal violet staining and colony-forming unit (CFU) enumeration. After biofilm formation on titanium discs, the samples were subjected to each treatment. Quantitative analysis revealed that all treated groups showed reductions in both biofilm biomass and viable bacterial counts compared to the untreated control. Notably, the combined treatment resulted in significantly greater reductions than either agent alone.

Although formal synergy quantification was not conducted, the observed improvements suggest a potentially additive or synergistic interaction, meriting further investigation. Additional studies are required to determine whether the interaction is truly synergistic or simply additive (Figure 2B,C).

### 2.3. Live/Dead Assay of Biofilms Treated with PVP-I and H_2_O_2_

A live/dead fluorescence assay was performed to assess the efficacy of PVP-I and H_2_O_2_, both individually and in combination, in disrupting mature biofilms. This experiment served as a confirmatory analysis following CFU quantification and provided visual evidence of bacterial viability within the biofilm matrix.

Compared to the untreated control, quantitative image analysis revealed a moderate reduction in viable biomass following individual treatments. The live/dead staining showed increased red fluorescence (dead cells) and decreased green fluorescence (live cells), indicating that both agents compromised bacterial viability. Notably, the combined treatment induced a markedly higher proportion of dead bacteria, indicating a more substantial reduction in biofilm viability.

These results provide visual support for the enhanced antibiofilm activity of the combined PVP-I and H_2_O_2_ treatment and highlight its potential for managing biofilm-associated infections (Figure 3).

### 2.4. Expression of Biofilm-Associated Genes Following PVP-I and H_2_O_2_ Treatment

Gene expression analysis was performed on *S. aureus* biofilms after full maturation and were subsequently treated with PVP-I and H_2_O_2_. As the treatment was applied to pre-formed biofilms, the observed transcriptional changes likely reflect effects on biofilm maintenance mechanisms, rather than early-stage biofilm synthesis.

Downregulation of the *icaA*, *icaB*, *icaD*, and *icaR* genes suggests that the antiseptics disrupted the synthesis and structural stability of the extracellular matrix, while reduced expression of *clfA* indicates a diminished ability of bacterial cell-to-cell adhesion. These findings support the hypothesis that antiseptic treatment not only physically disrupts biofilms but also interferes with gene-regulated pathways critical for biofilm persistence (Figure 4).

## 3. Discussion

Our findings highlight the potential of combining PVP-I and H_2_O_2_ to overcome the formidable resistance of mature *S. aureus* biofilms. In our model, the dual treatment led to significantly greater reductions in biofilm biomass and bacterial viability than either agent alone, suggesting more effective disruption of both the biofilm matrix and bacterial integrity. This observation is consistent with prior studies, which showed that biofilm-embedded bacteria can be up to 1000 times more resistant to antimicrobials than their planktonic counterparts [9,16]. Notably, our results support the hypothesis that combining agents with complementary mechanisms—oxidative stress induced by H_2_O_2_ and protein denaturation mediated by PVP-I—can provide superior efficacy over monotherapy. Given the high prevalence of biofilm-related infections in PJIs [17,18,19], these findings may inform intraoperative disinfection strategies.

In this study, the antiseptic treatments were applied after the biofilms had fully matured, allowing us to assess their effects on biofilm stability and maintenance rather than on initial formation. Compared to single-agent treatments, the combined application of PVP-I and H_2_O_2_ led to significantly greater reductions in crystal violet staining, CFU counts, and live cell viability under confocal microscopy. Specifically, the average bacterial load in the control group was 9.13 log_10_ CFU/cm^2^, reduced to 6.32 by PVP-I, 7.07 by H_2_O_2_, and further decreased to 5.11 log_10_ CFU/cm^2^ with the combination treatment—demonstrating a nearly 4-log reduction within a 5 min exposure window.

At the molecular level, quantitative PCR analysis revealed significant downregulation of key biofilm-associated genes, including *icaA*, *icaB*, *icaD*, *icaR*, and *clfA*. These genes play essential roles in poly-N-acetylglucosamine (PIA) production, biofilm maturation, and bacterial aggregation. While it is reasonable to assume that a reduction in viable bacterial cells could lead to decreased gene expression, the internal control gene *16S* rRNA remained consistently expressed across all groups. This suggests that total bacterial RNA was not substantially reduced, supporting the conclusion that the observed gene suppression likely results from targeted disruption of transcriptional regulation rather than being merely a consequence of cell death.

The complementary actions of PVP-I and H_2_O_2_ may explain the observed improvements. PVP-I disrupts bacterial proteins and membranes [20], while H_2_O_2_ produces reactive oxygen species that damage both cellular components and the biofilm matrix. The initial application of H_2_O_2_ may disrupt the biofilm architecture, thereby enhancing PVP-I penetration and its bactericidal effects [21]. Therefore, theoretically, the use of PVP-I after the H_2_O_2_-mediated disruption of the biofilm structure can kill more bacteria [22,23]. However, this mechanism remains speculative and should be confirmed through further mechanistic studies such as transcriptomic profiling or in vivo imaging.

Moreover, the treatment duration in this study “5 min per agent” was chosen to reflect practical constraints in surgical procedures, particularly during the management of PJIs. In clinical settings, surgeons have limited time to disinfect explanted prosthetic components during revision surgeries. This short exposure period was therefore selected to reflect real-world surgical conditions. While longer exposure times may enhance antibiofilm efficacy, future studies should investigate varied durations to better understand the time-dependent effects and to refine protocols for maximum clinical benefit.

Clinically, our findings support the use of combination antiseptics to manage biofilm-related infections. Incorporating agents with distinct yet complementary actions may enhance efficacy while reducing the likelihood of resistance. However, further optimization is required, particularly in terms of concentration, exposure duration, and biocompatibility.

A limitation of the present study is the lack of molecular identification methods, such as *16S* rRNA sequencing or *nuc* gene amplification, which, if applied, would allow more accurate confirmation that the bacteria in the biofilms are *S. aureus*. Although morphological and biochemical methods were employed, future studies will incorporate PCR-based assays to ensure greater specificity. Additionally, only a single commercial isolate was used in this study. We recognize the importance of evaluating a broader panel of *S. aureus* strains and including certified reference strains such as ATCC 25923 for quality control and validation.

Despite the significant reductions in CFU counts and biofilm biomass observed in the combination group, we did not perform formal synergy quantification (e.g., FICI or Bliss Independence models). Therefore, we refer to the observed effect as “enhanced” rather than definitively synergistic. The concentrations of PVP-I and H_2_O_2_ used were based on standard clinical applications and preliminary screening; however, we acknowledge the absence of cytotoxicity testing and dose–response analysis as limitations. Future work should include detailed exploration of concentration gradients, assessment of cytotoxicity, and validation in in vivo models to optimize therapeutic application and safety.

## 4. Materials and Methods

### 4.1. Bacterial Strains and Phenotypic Confirmation of S. aureus

The *S. aureus* strain used in this study was purchased from Biozoa Biological Supply (Product No. 155067, Seoul, Republic of Korea) and maintained under biosafety level 2 (BSL-2) laboratory conditions. Since the strain was commercially obtained and not isolated from human subjects, no IRB approval was required.

The strain was cultured in tryptic soy broth (TSB; BD, Franklin Lakes, NJ, USA) and incubated overnight at 37 °C with shaking at 200 rpm. Following incubation, the suspension was adjusted to a 1.0 McFarland standard and diluted 1:300 to reach ~1 × 10^6^ CFU/mL.

Phenotypic confirmation was performed using mannitol salt agar (MSA; Fushenbio, Shanghai, China). The strain formed yellow colonies indicative of mannitol fermentation. Gram staining showed Gram-positive cocci in grape-like clusters under light microscopy (1000× magnification). Additionally, biofilm architecture visualized by confocal laser scanning microscopy (CLSM) revealed dense and structured biofilm formation typical of *S. aureus*. Although no molecular identification (e.g., *nuc* gene PCR) was performed, the combination of morphological and phenotypic data supports accurate strain identification.

### 4.2. Biofilm Formation Assay

#### 4.2.1. Preliminary Biofilm Formation (96-Well Plate)

To assess biofilm-forming capacity, an overnight culture was adjusted to a 1.0 McFarland standard in TSB. Then, 200 μL of the suspension was added to each well of a 96-well polystyrene plate (Falcon, Corning Inc., New York, NY, USA) and incubated at 37 °C for 48 h under static conditions. Wells were washed gently with PBS, stained with 0.1% crystal violet for 30 min, then solubilized with 95% ethanol. Absorbance was measured at 570 nm using a microplate reader (Synergy HTX, BioTek, Winooski, VT, USA). Strong biofilm production was confirmed (Figure 2A).

#### 4.2.2. Biofilm Formation on Titanium Discs

To obtain an overnight culture of *S. aureus*, the bacterial cells were cultured in TSB supplemented with 0.2% glucose. The overnight cultures were prepared in a shaking incubator (VS-8480SF, Vision Scientific Co., Ltd., Bucheon, Republic of Korea) at 37 °C and 200 rpm. After adjusting the bacterial suspension, titanium alloy metal discs (approximately 8 mm in diameter and 1 mm in thickness), which were obtained from the screw holes of acetabular cups provided by Lima Corporate (Villanova di San Daniele del Friuli, Italy), were placed in a 24-well tissue culture plate (Falcon, Corning Inc., New York, NY, USA). Bacterial suspension was added to each well until the total volume reached 2 mL, which corresponds to the full capacity of a 24-well plate well. This ensured that the metal discs were completely submerged, minimizing the air–liquid interface and better simulating in vivo conditions. The plates were incubated at 37 °C with 5% CO_2_ for 48 h in a CO_2_ incubator (MCO-175, Panasonic, Osaka, Japan) [24].

Biofilm formation was confirmed by crystal violet staining and confocal laser scanning microscopy (CLSM; LSM900 with Airyscan2, ZEISS, Oberkochen, Germany) using a live/dead bacterial viability assay, as detailed in later sections.

### 4.3. Application of Treatments

After incubation for 48 h, biofilms were allowed to form on the surface of the metal discs. These pre-formed biofilms were then subjected to various treatments to evaluate the agents’ ability to disrupt mature biofilm structures. The discs were divided into four groups: untreated control (sterile phosphate-buffered saline, PBS), 1% PVP-I, 3.5% H_2_O_2_, and a combination treatment. PBS was used as a negative control to account for the effects of washing and handling. For the combination group, discs were first treated with 3.5% H_2_O_2_ for 5 min, followed by 1% PVP-I for 5 min (Table 1).

### 4.4. CFU Enumeration

Following treatment, the metal discs were placed in 1 mL of sterile PBS and subjected to ultrasonic treatment at 35 kHz for 15 min, with shaking for 10 s every 5 min. Serial 10-fold dilutions were prepared, and 50 μL of the appropriately diluted samples were plated on MSA (mannitol salt agar; NutriSelect Basic, Millipore, Darmstadt, Germany). The plates were incubated at 37 °C for 24 h, and colony-forming counts (CFUs) were counted. The CFU enumeration assay was independently repeated seven times.

### 4.5. Biofilm Detection

After treatment with PVP-I, H_2_O_2_, and PBS, the wells were washed twice with PBS to remove any loosely attached bacteria. For biofilm staining, each metal disc was stained with 1 mL of 0.1% crystal violet solution. The plates were then incubated at room temperature (~25 °C) for 25 min. After staining, the dye was removed, and the discs were washed thrice with PBS and dried at 37 °C. The biofilm was then dissolved in 1 mL of 95% ethanol. Absorbance was measured at λ = 570 nm using a microplate reader. This assay was repeated eight times [25].

### 4.6. Live/Dead Assay

Biofilms from control and treatment groups were imaged using confocal microscopy to verify the CFU data and to determine whether viable bacteria remained embedded within the biofilm structures after treatment of the biofilm. The SYTO-9/PI Live/Dead BacLight Bacterial Viability Kit (Fushenbio, Shanghai, China) was used to visualize bacterial biofilms, according to the manufacturer’s instructions. Live bacteria are stained green by SYTO-9, while dead cells are stained red because of the uptake of propidium iodide (PI) through compromised cell membranes. After treatment, the biofilm samples were gently washed with sterile water to remove loosely attached bacteria and debris. The samples were then stained in the dark at room temperature (~25 °C) for 15 min, washed with PBS, and imaged using a Zeiss LSM900 confocal microscope equipped with Airyscan2 (ZEISS, Oberkochen, Germany) [25]. All imaging experiments were performed in triplicate. Confocal images were analyzed using Fiji (ImageJ, version 1.52p; National Institutes of Health, Bethesda, MD, USA).

### 4.7. Gene Expression Analysis (RT-PCR)

Previous studies have linked the biofilm inhibitory activity of PVP-I to the decreased transcription of the *icaADBC* operon and the induction of *icaA* transcription repression in *S. aureus* [26,27,28,29]. To perform PCR, total RNA was extracted, and 0.5 μg of total RNA was used to synthesize first-strand cDNA in a 20 μL reaction mixture containing 200 U M-MLV reverse transcriptase, 0.25 mM DTT, and 250 μM each of dATP, dCTP, dGTP, and dTTP. The reverse transcription PCR conditions were as follows: initial incubation at room temperature (~25 °C) for 10 min, followed by 12 cycles at 25 °C for 30 s, 45 °C for 4 min, and a gene-specific annealing step (temperatures listed in Table 2) for 30 s, and a final step of 5 min heating at 95 °C and storage at 4 °C. The cDNA was then amplified using AccuPower GreenStar PCR PreMix (Bioneer, Daejeon, Republic of Korea) in a MyGenie 96/384 thermal cycler. Gene expression analysis was performed targeting *clfA*, *icaA*, *icaB*, *icaD*, *icaR*, and the *16S* rRNA gene, with the *16S* rRNA gene used as the housekeeping gene for normalization and assessment (Bioneer, Daejeon, Republic of Korea) [30,31]. All reactions were performed in triplicate. The oligonucleotide sequences used for gene expression analysis are listed in Table 2 [7].

### 4.8. Statistical Analysis

Statistical analyses were performed using GraphPad Prism 9.0 (GraphPad Software Inc., San Diego, CA, USA). Results are expressed as the mean ± standard deviation (SD). Preliminary biofilm formation assays (OD570 in 96-well plates) were repeated eight times and analyzed using the Mann–Whitney U test due to the small sample size and non-normal distribution. Treatment-related OD570 biofilm assays were repeated eight times, CFU enumeration seven times, and gene expression and CLSM assays were each performed in triplicate. Differences between treatment groups were analyzed using one-way ANOVA followed by Tukey’s post hoc test. *p* < 0.05 was considered statistically significant.

## 5. Conclusions

This study demonstrated that both PVP-I and H_2_O_2_, individually and in combination, can effectively disrupt pre-formed *S. aureus* biofilms. The combined application of these two agents resulted in greater reductions in biofilm biomass and gene expression than either agent alone. While formal synergy quantification was not performed, the enhanced effect observed suggests a potentially additive or synergistic interaction. These findings support the potential utility of employing dual-antiseptic strategies with complementary mechanisms of action to enhance biofilm control, particularly in medical device-associated infections. Further in vivo and clinical investigations are warranted to validate these findings and to assess the long-term safety, optimal dosing, and therapeutic potential of this dual-agent strategy.

## Figures and Tables

**Figure 1 ijms-26-04390-f001:**
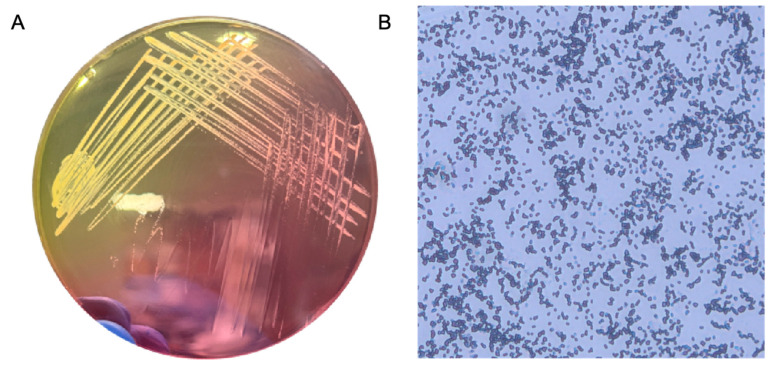
Morphological and culture-based identification of *S. aureus.* (**A**) Colonies of *S. aureus* grown on mannitol salt agar (MSA), showing characteristic yellow pigmentation due to mannitol fermentation. (**B**) Gram-stained image of *S. aureus* observed under light microscopy, displaying purple-colored cocci arranged in grape-like clusters.

**Figure 2 ijms-26-04390-f002:**
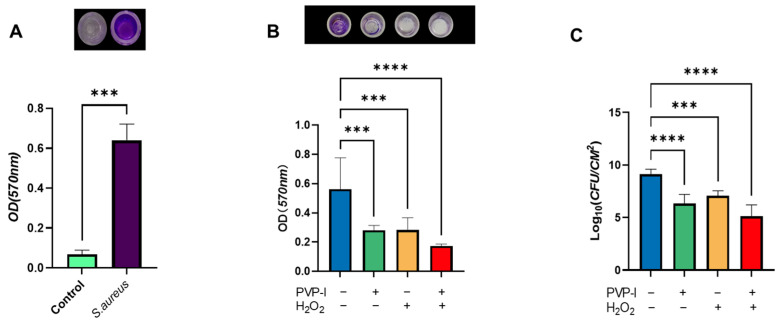
Evaluation of *S. aureus* biofilm formation and disruption after treatment with PVP-I and H_2_O_2_. (**A**) Initial biofilm-forming ability of *S. aureus* was quantified using crystal violet staining at 570 nm. Data are presented as mean ± standard deviation *(n* = 8). Statistical significance was assessed using the Mann–Whitney U test. *** *p* < 0.001 versus control. (**B**) Quantification of biofilm biomass using crystal violet staining at 570 nm. Data are presented as mean ± standard deviation (*n* = 8). (**C**) Viable bacterial cells (log_10_ CFU/cm^2^) after treatment. Data are presented as the mean ± standard deviation (*n* = 7). Statistical significance was assessed using one-way ANOVA followed by Tukey’s post hoc test. *** *p* < 0.001, **** *p* < 0.0001 compared with the control group.

**Figure 3 ijms-26-04390-f003:**
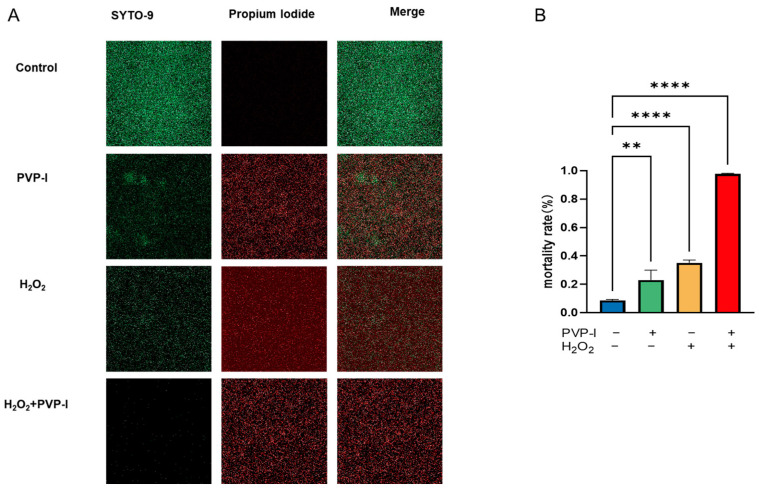
Confocal microscopy images from the live/dead assay demonstrating reduced viability in *S. aureus* biofilms treated with PVP-I and H_2_O_2_. (**A**) Representative fluorescence images of pre-formed *S. aureus* biofilms stained with SYTO 9 (live cells, green) and propidium iodide (dead cells, red), captured using confocal laser scanning microscopy at 40× magnification; scale bar = 100 µm. (**B**) Quantification of the live/dead cell ratio calculated using Fiji software (ImageJ, version 1.52p). Data are presented as mean ± SD (*n* = 3). Statistical significance was evaluated using one-way ANOVA followed by Tukey’s post hoc test. ** *p* < 0.01, and **** *p* < 0.0001 versus the control group.

**Figure 4 ijms-26-04390-f004:**
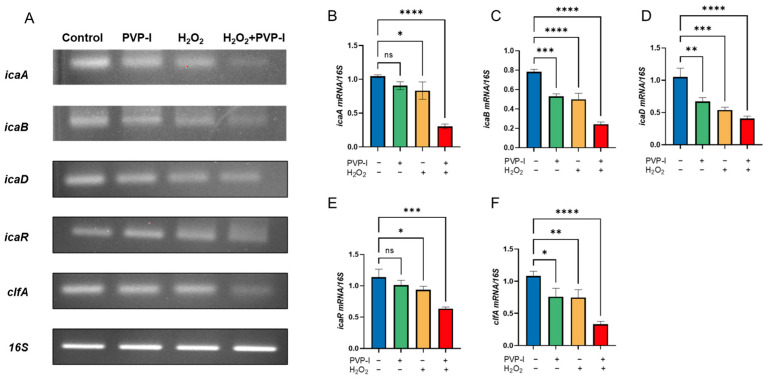
(**A**) RT–qPCR analysis showing reduced expression of biofilm-associated genes (*icaA*, *icaB*, *icaD*, *icaR*, and *clfA*) in the PVP-I, H_2_O_2_, and combination treatment groups. (**B**–**F**) Quantitative expression levels of each gene normalized to *16S* rRNA: *icaA* (**B**), *icaB* (**C**), *icaD* (**D**), *icaR* (**E**), and *clfA* (**F**). Notably, *16S* rRNA expression remained stable across all groups, indicating that the downregulation of biofilm-related genes was not solely due to bacterial death but may reflect regulatory interference induced by the treatments. Data are presented as mean ± standard deviation (*n* = 3). Statistical analysis was performed using one-way ANOVA followed by Tukey’s post hoc test. ns = not significant, * *p* < 0.05, ** *p* < 0.01, *** *p* < 0.001, **** *p* < 0.0001 compared to the control group.

**Table 1 ijms-26-04390-t001:** Treatment Conditions Applied to Biofilm Samples.

Treatment Group	Agent(s) Used	Concentration	Exposure Time	Treatment Order
Control	PBS	—	—	—
PVP-I	Povidone-iodine	1%	5 min	Single treatment
H_2_O_2_	Hydrogen peroxide	3.5%	5 min	Single treatment
Combination	H_2_O_2_ + PVP-I	3.5% + 1%	5 min + 5 min	H_2_O_2_ first, then PVP-I

**Table 2 ijms-26-04390-t002:** Primers and annealing temperatures for gene amplification.

Gene	Sequence (5′–3′)	Annealing Temp (°C)
*icaA*	F: CTATTTCGGGTGTCTTCACTC	53.7
	R: GGCAAGCGGTTCATACTTA	
*icaB*	F: TTGCCTGTAAGCACACTGGATGGTC	57.5
	R: TACACGGTGATAATTTAATGCCAGAGC	
*icaD*	F: ATGGACAAGTCCAGACAGAGGAAAA	59.1
	R: GTCACTCATCGTAACTGCTTCAACG	
*icaR*	F: TCAGAGAAGGGGTATGACGGTACAA	58.2
	R: TCCTCAGGCGTATTAGATAATTGAACG	
*clfA*	F: CAAGTAGCGTTAGTGCTGC	51.8
	R: TGATTGAGTTGTTGCCG	
*16S*	F: CGTGCTACAATGGACAATACAAA	57.2
	R: ATCTACGATTACTAGCGATTCCA	

## Data Availability

All data generated or analyzed in this study are included in the article.

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
