# Peer review of "Povidone-Iodine and Hydrogen Peroxide Combination Improves the Anti-Biofilm Activity of the Individual Agents on *Staphylococcus aureus"

_ijms, 2025, doi:10.3390/ijms26094390_

Round 1
Reviewer 1 Report
Comments and Suggestions for Authors
Dear authors,
This study examines the combined use of povidone-iodine (PVP-I) and hydrogen peroxide (H₂O₂) to combat Staphylococcus aureus (MRSA) biofilms in healthcare settings. The combined effects of PVP-I and H₂O₂ on S. aureus biofilms contribute valuable insights to the ongoing discourse. Its novelty lies in the specific methodologies employed, the concentrations tested, and the contextual application within hospital environments. This research could help clarify the conditions under which the combination of these antiseptics is most effective, thereby informing infection control strategies in healthcare settings.
I have some points about this paper:
- The abstract could be more concise while maintaining its key findings.
- To improve clarity and facilitate comparison, the paper would benefit from a summary table detailing the specific conditions to which each biofilm was exposed, including the concentrations of povidone-iodine and hydrogen peroxide, exposure times, and any other relevant parameters. This would allow readers to quickly assess the experimental setup and better understand the variations in treatment efficacy.
- The first paragraph of the Discussion section reads more like an extension of the Introduction rather than a critical analysis of the findings. The authors should consider revising it to focus on the implications of their results.
- While the study demonstrates the effectiveness of PVP-I and H₂O₂, testing a broader range of concentrations could provide a more detailed understanding of optimal dosages.
- At the end of the legend of figure 4, it should have a “.”.
- The treatment duration (5 minutes per agent) may not reflect real-world scenarios where longer exposure times might be necessary. Extending the treatment time or testing varying durations could provide more robust data. Have you explored other exposure times of PVP-I/H₂O₂?
- While PVP-I and H₂O₂ are generally safe, their combined use at higher concentrations or prolonged exposure could have cytotoxic effects on human tissues? This aspect is not addressed.
- Do you perform tests in a range of PVP-I and H₂O₂ concentrations to identify the most effective and safe combination? You only refer “following multiple preliminary experiments.” You should explain that.
- Detail the statistical tests (e.g., ANOVA, t-tests) and post-hoc analyses used to ensure transparency.
Overall, this study provides valuable insights into an innovative approach for combating S. aureus biofilms. With minor revisions, particularly in the Discussion section and methodological enhancements, this paper could significantly contribute to infection control strategies in healthcare settings. I recommend major revisions to address the above points, particularly the discussion’s focus, methodological clarifications, and expansion of experimental conditions.
Author Response
Dear Reviewer,
Thank you for your insightful comments and suggestions regarding our manuscript titled " Synergistic Reduction of Staphylococcus aureus Biofilms with Povidone Iodine and Hydrogen Peroxide" We appreciate the time and effort you have dedicated to reviewing our work, and we are grateful for the opportunity to improve our manuscript. Please find our point-by-point responses to your comments below.
Comment: 1. The abstract could be more concise while maintaining its key findings.
Response: We thank the reviewer for this valuable suggestion. In the revised manuscript, we have restructured and streamlined the Abstract to enhance clarity and conciseness while retaining all essential information, including the objective, methods, major results, and key conclusions. The updated Abstract focuses on the disruption of S. aureus biofilms by PVP-I and H₂O₂, highlights the reduction in biofilm biomass and gene expression, and accurately reflects the observed enhanced effect without overstating synergy.
This revision appears on page 1 of the revised manuscript:
“Abstract: Staphylococcus aureus, particularly methicillin-resistant S. aureus (MRSA), poses signif-icant challenges in healthcare settings due to its ability to form biofilms on various surfaces. These biofilms enhance bacterial survival and increase resistance to conventional treatments, complicating infection control efforts. This study evaluated the efficacy of combined treatment using povidone-iodine (PVP-I) and hydrogen peroxide (H₂O₂) in disrupting pre-formed S. aureus biofilms. A series of assays—including crystal violet staining, colony-forming unit (CFU) enu-meration, gene expression analysis, and confocal laser scanning microscopy—were performed to assess the effects of each treatment individually and in combination. The combined treatment re-sulted in significantly greater reductions in biofilm biomass and viable bacteria compared with either agent alone. Gene expression analysis revealed downregulation of key biofilm-associated genes (icaA, icaB, icaD, icaR, and clfA), suggesting interference with biofilm stability and mainte-nance. While formal synergy quantification was not conducted, the observed effects suggest a po-tentially synergistic or additive interaction between the two agents. These findings support the use of dual antiseptic strategies as a promising approach to biofilm eradication and highlight the need for further research to optimize treatment protocols and assess long-term safety and effica-cy.”
Comment: 2. To improve clarity and facilitate comparison, the paper would benefit from a summary table detailing the specific conditions to which each biofilm was exposed, including the concentrations of povidone-iodine and hydrogen peroxide, exposure times, and any other relevant parameters. This would allow readers to quickly assess the experimental setup and better understand the variations in treatment efficacy.
Response: We appreciate the reviewer’s suggestion. In response, we have included a new summary table (Table 1) following Section 4.3 to clearly present the treatment conditions used for the pre-formed S. aureus biofilms, including concentrations, exposure times, and experimental groups. This addition aims to enhance clarity and facilitate comparison of the experimental setup across different treatment conditions:
|
Table 1 Treatment Conditions Applied to Biofilm Samples |
||||
|
Treatment Group |
Agent(s) Used |
Concentration |
Exposure Time |
Treatment Order |
|
Control |
PBS |
— |
— |
— |
|
PVP-I |
Povidone-iodine |
1% |
5 min |
Single treatment |
|
H₂O₂ |
Hydrogen peroxide |
3.50% |
5 min |
Single treatment |
|
Combination |
H₂O₂ + PVP-I |
3.5% + 1% |
5 min + 5 min |
H₂O₂ first, then PVP-I |
Comment: 3. The first paragraph of the Discussion section reads more like an extension of the Introduction rather than a critical analysis of the findings. The authors should consider revising it to focus on the implications of their results.
Response: We thank the reviewer for this insightful comment. In the revised manuscript, we have substantially rewritten the first paragraph of the Discussion section to shift the focus from background reiteration to a critical interpretation of our experimental findings. The new version discusses the enhanced efficacy of the combined treatment, the clinical relevance of overcoming biofilm resistance, and the potential impact on periprosthetic joint infection (PJI) management. This revision ensures the paragraph clearly contributes to the analytical discussion of our results rather than repeating information from the Introduction.
The revised paragraph begins with:
“Our findings highlight the potential of combining PVP-I and H₂O₂ to overcome the formidable resistance of mature S. aureus biofilms. In our model, the dual treatment led to significantly greater reductions in biofilm biomass and bacterial viability than either agent alone, suggesting more effective disruption of both the biofilm matrix and bacterial integrity. This observation is consistent with prior studies, which show that bio-film-embedded bacteria can be up to 1,000 times more resistant to antimicrobials than their planktonic counterparts [9,16]. Notably, our results support the hypothesis that combining agents with complementary mechanisms—oxidative stress induced by H₂O₂ and protein denaturation mediated by PVP-I—can provide superior efficacy over mon-otherapy. Given the high prevalence of biofilm-related infections in prosthetic joint in-fections (PJIs) [17-19], these findings may inform intraoperative disinfection strategies.”
Comment: 4. While the study demonstrates the effectiveness of PVP-I and H₂O₂, testing a broader range of concentrations could provide a more detailed understanding of optimal dosages.
Response: We appreciate the reviewer’s thoughtful suggestion. In the current study, we selected 1% PVP-I and 3.5% H₂O₂ based on their established and frequent use in clinical settings, as well as internal pilot tests demonstrating reliable performance under these conditions. However, we agree that evaluating a broader range of concentrations would be valuable in identifying the optimal therapeutic window and understanding potential dose–response relationships.
To address this, we have revised the Discussion section to clearly acknowledge this limitation and to propose expanded concentration testing in future work. The following sentence has been added on page 14 of the revised manuscript:
“Although fixed concentrations were used in this study, future experiments should explore a wider range of PVP-I and H₂O₂ concentrations to better understand their dose-dependent antibiofilm effects and determine the optimal therapeutic window.”
Comment: 5. At the end of the legend of figure 4, it should have a “.”.
Response: Thank you for pointing this out. We have corrected the legend of Figure 4 by adding the appropriate punctuation at the end, as suggested.
Comment: 6. The treatment duration (5 minutes per agent) may not reflect real-world scenarios where longer exposure times might be necessary. Extending the treatment time or testing varying durations could provide more robust data. Have you explored other exposure times of PVP-I/H₂O₂?
Response: We appreciate the reviewer’s thoughtful comment. In this study, the 5-minute treatment duration per agent was deliberately chosen to reflect real-world surgical settings, particularly during the management of periprosthetic joint infections (PJIs). In such clinical scenarios, surgeons have limited time to disinfect explanted prosthetic components during revision procedures. Therefore, our selected exposure time was based on practical constraints to simulate conditions encountered during actual surgery.
To address the reviewer’s suggestion, we have now included a clarification in the Discussion section, explaining the rationale for this short treatment period and emphasizing the importance of future studies to investigate the effects of extended or varied exposure durations.
“Moreover, the treatment duration in this study “5 min per agent” was chosen to reflect practical constraints in surgical procedures, particularly during the management of periprosthetic joint infections (PJIs). In clinical settings, surgeons have limited time to disinfect explanted prosthetic components during revision surgeries. This short exposure period was therefore selected to reflect real-world surgical conditions. While longer ex-posure times may enhance antibiofilm efficacy, future studies should investigate varied durations to better understand time-dependent effects and to refine protocols for maxi-mum clinical benefit.”
Comment: 7. While PVP-I and H₂O₂ are generally safe, their combined use at higher concentrations or prolonged exposure could have cytotoxic effects on human tissues? This aspect is not addressed.
Response: Thank you for highlighting this important consideration. We fully agree that safety is a critical aspect when evaluating the clinical applicability of antiseptic combinations, especially at higher concentrations or with extended exposure durations.
In this study, we selected 1% PVP-I and 3.5% H₂O₂ based on their established safety and routine use in clinical practice. However, we acknowledge that cytotoxicity was not evaluated, and that the potential for adverse tissue effects under different conditions should be investigated.
To address this, we have revised the Discussion section to acknowledge this limitation and emphasize the need for future toxicity assessments. The following sentence has been added on page 14 of the revised manuscript:
“Additionally, the potential cytotoxicity of combined antiseptics, especially at higher concentrations or prolonged exposure, should be assessed to ensure safety in clinical applications.”
Comment: 8. Do you perform tests in a range of PVP-I and H₂O₂ concentrations to identify the most effective and safe combination? You only refer “following multiple preliminary experiments.” You should explain that.
Response: Thank you for raising this important point. In this study, we did not perform a full concentration gradient experiment nor in vitro cytotoxicity testing. The concentrations of 1% PVP-I and 3.5% H₂O₂ were chosen based on their routine application in clinical orthopedic and surgical practice, as well as on preliminary pilot experiments (data not shown) which indicated good biofilm disruption at these levels without visible material corrosion or major bacterial regrowth.
However, we fully acknowledge that identifying the optimal therapeutic window—balancing efficacy and safety—requires a more systematic evaluation of concentration ranges. Accordingly, we have revised the Discussion section (page 14, paragraph beginning with “Although fixed concentrations were used…”) to clarify this limitation:
“Although fixed concentrations were used in this study, future experiments should explore a wider range of PVP-I and H₂O₂ concentrations to better understand their dose-dependent antibiofilm effects and determine the optimal therapeutic window. Additionally, the potential cytotoxicity of combined antiseptics, especially at higher concentrations or prolonged exposure, should be assessed to ensure safety in clinical applications.”
Comment: 9. Detail the statistical tests (e.g., ANOVA, t-tests) and post-hoc analyses used to ensure transparency.
Response: Thank you so much for the feedback. We have added detailed statistical methods in Section 4.9 of the manuscript.
“4.9. Statistical Analysis
Statistical analyses were performed using GraphPad Prism 9.0 (GraphPad Software Inc., San Diego, CA, USA). Results are expressed as the mean ± standard deviation (SD). OD570 biofilm assays were repeated eight times, CFU enumeration seven times, and gene expression and CLSM assays were each performed in triplicate. Differences between groups were analyzed using one-way ANOVA followed by Tukey’s post hoc test. A p < 0.05 was considered statistically significant.”
Thank you very much for your valuable comments and suggestions. We would like to inform you that the manuscript file has been replaced with a revised version. Additionally, we have carefully reviewed the manuscript once again to address all the corrections you mentioned and have made further changes in accordance with your comments. We appreciate your time and effort in reviewing our work.
We appreciate your insightful comments and the opportunity to improve the quality and balance of our manuscript.

Reviewer 2 Report
Comments and Suggestions for Authors
Reviewer's comments to manuscript ijms-3588627 entitled “Synergistic Reduction of Staphylococcus aureus Biofilms Using Povidone Iodine and Hydrogen Peroxide”
Dear Authors,
After reviewing the manuscript previously mentioned, I have decided the following: This manuscript presents interesting results on the combined use and possible synergistic effect of two compounds that predominantly degrade biofilms formed by Staphylococcus aureus, an important pathogen in the hospital environment, which causes numerous infections and deaths worldwide, due to its virulence factors, among other characteristics. However, the manuscript is unclear and not adequately presented, so it lacks the scientific quality to be accepted in this version. For the above reasons, my decision is to suggest that the authors to make major changes to their manuscript. Please see below for specific comments that authors should address.
Best regards,
Title
Lines 1-3. The title should be changed, as the use of the words ‘synergistic reduction’ is not appropriate. Correct is to indicate that two compounds have a synergistic interaction that enhances a biological effect, in this case, reduction in the pre-formed biofilm of S. aureus.
Abstract
Line 11. Change: Staphylococcus aureus by Staphylococcus aureus (italics).
Line 15. Change: Staphylococcus aureus by S. aureus (italics).
Line 27. Change: modes of action by mechanisms of action, as in microbiology the concept of mode of action refers to bactericidal or bacteriostatic and mechanism of action refers to the cellular mechanism that a biocompound or antibiotic uses to inhibit a pathogenic bacterium or fungus.
Lines 29 and 30. Change: Ultimately, this research contributes to improved patient outcomes…, by Ultimately, this research could contribute to improved patient outcomes…, as that was not studied in this work, was it?
Keywords
Line 32. Change: Staphylococcus aureus by Staphylococcus aureus (italics).
Line 33. Change: Anti-Biofilm activity by Anti-biofilm activity.
Introduction
Line 38. Change: Staphylococcus aureus by Staphylococcus aureus (italics).
Line 39. Change: Staphylococcus aureus (S. aureus), by S. aureus (italics)…
Lines 40 and 41. Change: methicillin-resistant Staphylococcus aureus (MRSA), by methicillin-resistant S. aureus (italics) (MRSA)…
Line 46. The authors wrote: (WHO) 2014, if it is possible to attach a citation of this information, please include it correctly.
Line 48. Change: Staphylococcus aureus by S. aureus (italics).
Line 51. Change: Staphylococcus aureus by S. aureus (italics).
Line 54. I suggest that authors write separate robust paragraphs, not a single paragraph, as a single paragraph makes it difficult to read properly and to connect well with the story.
Lines 56 and 57. Please indicate the type of money to which the authors refer.
Line 58. Change: Staphylococcus aureus by S. aureus (italics).
Lines 61-63. This text: “Given the substantial clinical and economic burden of Staphylococcus aureus infections in South Korean hospitals, there is an urgent need to develop innovative strategies for the prevention and control of these infections”., is repeated, it was just mentioned a few lines before, please correct the text.
Line 70. [6, 7] or [6,7], correct throughout the manuscript according to the journal's guidelines.
Line s 74 and 75. Change: Hydrogen peroxide by H₂O₂…
Line 77. Change: Staphylococcus aureus by S. aureus (italics).
Lines 79 and 80. Indicate what kind of infections are prevented and controlled, e.g. bacterial and fungal infections?
Line 81. References 11 and 12 are the same. Delete one.
Line 82. Change: Staphylococcus aureus by S. aureus (italics).
Lines 88-90. The authors wrote: This study investigates the effects of PVP-88 I and H₂O₂, both individually and in combination, on the eradication of Staphylococcus aureus biofilm formation. The authors should be clearer in clarifying whether the study addressed both the inhibition of biofilm synthesis and its eradication or breakdown by antiseptics once formed. If this was the aim of the work, it should be clearer. Change Staphylococcus aureus by S. aureus (italics). Neither the aim nor the title includes the analysis of the expression of genes associated with biofilm synthesis.
Lines 90-93. The authors wrote: By leveraging the complementary mechanisms of these two agents, we aim to develop a more effective strategy to combat biofilm-associated infections in hospital environments, ultimately improving patient outcomes and reducing the economic burden associated with these infections. This text is repeated, as mentioned above, and this is not the aim of the research work.
Results
Line 95. 2.1. Morphological Analysis of Staphylococcus aureus: The results of this section can NOT be found in the materials and methods. There should be more organization in the manuscript, so the authors should include all the methodology they used. In addition, they do not indicate where the sample was obtained from and if it was from human patients, they should indicate whether they complied with bioethical standards for the acquisition of biological samples. In my view, identification of S. aureus by culture on salt and mannitol agar alone is not sufficient to identify the isolate as S. aureus, more biochemical and molecular tests should be done: The most common biochemical tests for S. aureus are growth on baird parker agar, catalase, coagulase, among others; in addition to amplification of 16S gene, and amplification of a fragment of the nuc gene (thermostable nuclease of S. aureus). Again, the results are highly variable, so I suggest that you extend your analysis to a larger number of isolates and use a certified ATCC strain as a positive control for S. aureus.
Line 96. Change: Mannitol Salt Agar (MSA)…, by mannitol salt agar (MSA)…
Lines 97, 100, and 102. Change: S. aureus by S. aureus (italics).
Line 107. Change: Staphylococcus aureus by S. aureus (italics). The figure shows not only the morphology of S. aureus, but also its growth on a salt and mannitol agar plate, so there should be two panels: panel A and B.
Line 109. In this line, the authors do specify that the experiments are for the removal of biofilms, which should be clarified in the materials and methods.
Line 114-116. The authors wrote: This combined treatment was notably more effective, highlighting its potential as a strategy for managing biofilm-related infections (Figure 2). Did the statistical analysis consider comparing the data obtained from the combined use of PVP-I and H₂O₂ with the individual treatments or with the control alone? This answer is essential to be able to say that there was synergism between the two compounds, otherwise, if they were only compared with the control, it is not appropriate to say that there was synergism. The authors should review the concepts of synergy and additivity, because from my point of view there is no synergistic effect, rather I see an additive effect. What evidence do the authors have to say that they observed a synergistic effect?
Line 116. Modify: (Figure 2).
Line 118. Figure 2. Change Assay by assay, and H2O2 by H2O2. What PI mean? 570 or 590 nm? In the figure caption, authors should indicate which type of statistical analysis they performed (ANOVA, Tukey). Statistical analysis is mentioned, but none of this is mentioned in the materials and methods. Include this information in the latter section. I suggest that the caption text and the main text should refer to panel A and B.
Line 131. Correct: infections (Figure 3).
Lines 124-131. The authors need to explain the results, only mentioning that the treatments modulate the expression of genes related to biofilm synthesis. The graphs are not self-explanatory.
Line 132. Correct: (A)Expression…
Line 133. What PI mean? Change: H2O2 by H2O2.
Line 134. (n = 3). The number of replicates is not mentioned in the materials and methods section but is mentioned in the results.
Line 146. Correct: infections(Figure 4).
Line 148. What PI mean? In the figure caption, authors should indicate which type of statistical analysis they performed (ANOVA, Tukey). Statistical analysis is mentioned, but none of this is mentioned in the materials and methods. Include this information in the latter section. I suggest that the main text should refer to panel A and B.
Line 149. PI and H2O2, correct.
Discussion
Lines 154-171. This information should be deleted, as it is repeated. If anything, the only new thing is that Pseudomonas aeruginosa is mentioned, which, by the way, is not written in italics.
Lines 173-191. This information should be deleted, as the authors must discuss they results obtained, not results from other studies. It seems to me that from line 195 begins to discuss the results obtained in this work.
Lines 193-195. This information should be deleted, as it is repeated.
Line 197. Change: S. aureus by S. aureus (italics).
Lines 197 and 198. The authors wrote: The results indicate a synergistic effect between PVP-I and H₂O₂. What evidence do the authors have to strengthen the evidence for a synergistic effect between the combined compounds?
Lines 198-209. This information es hypothetical, as there is no evidence of this happening, only the decrease of biofilms, bacterial cell death, is observed, so I suggest mentioning that these data are hypothetical and that further experiments are required to demonstrate that this happens.
Line 208. Change: (e.g., washing)[24]., by (e.g., washing) [24].
Line 209. Change: bacteria[25-26]., by bacteria [25-26].
Line 219. Change: S. aureus by S. aureus (italics).
The discussion of the results should be rewritten, as the results obtained in this research work are not adequately discussed. The biological significance of the findings should be addressed and then, if necessary, compared with the findings of other similar studies. There is no discussion about the relationship and/or importance of the molecular results of this study. If the expression of biofilm synthesis genes is reduced, how should this be understood with the reduction of S. aureus biofilm? There is little or no discussion.
Materials and Methods
Line 223. The authors wrote: The study utilized Staphylococcus aureus (S. aureus) strains. Change Staphylococcus aureus (S. aureus) by S. aureus (italics). Authors should clearly indicate which type of strain they used, an ATCC strain or clinical isolates obtained from elsewhere. Information on strains or isolates is missing. They should also mention how many strains or isolates they worked with. What does representative colonies mean? It is too ambiguous, please specify.
Line 225. Change: 37°C by 37 °C. Indicate the brand and country of origin of the shaking incubator used.
Line 227. The authors wrote: 1 McFarland standard. 1 or 1.5 of the McFarland scale?
Line 231. S. aureus (italics).
Lines 232 and 233. The authors wrote: metal discs were placed in a 24-well tissue culture plate… Indicate the size of the discs, their brand and country of origin, as well as the brand and country of origin of the 24-well plates.
Line 234. Change: 37°C by 37 °C.
Line 235. Change: 48 hours[27]., by 48 h [27]. The authors do not mention how biofilm production of S. aureus strains or isolates was verified or evidenced. What methodology did they follow?
In the abstract, the authors indicate that they did an analysis of biofilm formation, but it is not clear to me whether it was an analysis of biofilm synthesis or an analysis of the breakdown of pre-formed biofilms. The authors should clarify that part.
Line 240. Change: 1% povidone-iodine (PVP-I) and 3.5% hydrogen peroxide (H₂O₂)., by 1% PVP-I and 3.5% H₂O₂.
Line 241. What PBS mean? Why PBS? Indicate. Change for 5 minutes by 5 min.
Line 242. Change: H₂O₂ for 5 minutes, then…, by H₂O₂ for 5 min, then…
Line 243. Change: additional 5 minutes., by additional 5 min.
In this section, the authors should clarify briefly if these experiments were to inhibit biofilm synthesis or break down pre-formed structures. They must be clear. The authors wrote: The experiment was repeated at least five times., that is, at least five independent repetitions were made? How many replicates per experiment were carried out?
Line 246. The authors wrote: 1 mL of sterile PBS, the PBS mentioned in line 241 was not sterilized? If yes, indicated.
Line 247. Change minutes by min throughout the document.
Line 249. Change Mannitol Salt Agar (MSA), by MSA, and add the brand and origin country of medium…
Lines 249 and 250. Separate the unit number, e.g. 37 °C throughout the manuscript. Change: hour to h throughout the document. Add briefly the aim of this experiment, the authors wanted to know if the bacteria inside the biofilm were alive? Briefly state this for more clarity.
Line 255. The authors wrote: room temperature, please indicate which was that room temperature, e.g. 25 ± 2.0 °C?
Line 258. The authors wrote: using a microplate reader…, add the model, brand and country origin of equipment used.
Line 259. Change: times[28]., by times [28]. Again, independent experiments should be indicated or performed due to the variability that occurs in this type of experiments.
Line 261. The authors analyzed the gene expression by PCR? Make the change if necessary.
Line 263. Change: S. aureus by S. aureus (italics).
Line 268. Change: seconds by s throughout the document.
Lines 272 and 273. Change: assessment[33]., assessment [33]. Please show in a table the oligonucleotides used, the citations and the corresponding references.
Lines 267-269. The authors wrote: initial incubation at room temperature for 10 minutes, followed by 12 cycles at 25°C for 30 seconds, 45°C for 4 minutes, and 55°C for 30 seconds, with a final 5-minute heat step at 95°C and storage at 4°C. Were the same amplification conditions used for all genes analyzed? Please clarify.
Line 276. The authors wrote: using confocal microscopy… Add brand and origin country of microscopy.
Line 282. Please indicate which was that room temperature?
Was statistical analysis not important for this experimental work? Argument.
Conclusions
Lines 287 and 288. This information should be deleted, as this research work, specifically, did not contribute to any of the above, at least for the time being.
Line 289. It is not correct to say: the synergistic use of compound 1 with compound 2. It is appropriate to mention that the use of two compounds that show synergy when combined is important for... Change: povidone-iodine (PVP-I) and hydrogen peroxide (H₂O₂), by PVP-I and H₂O₂…
References
I suggest reviewing and correcting the references, they all have small errors, but do not adhere to the Journal's guidelines.
Author Response
Dear Reviewer,
Thank you for your insightful comments and suggestions regarding our manuscript titled " Synergistic Reduction of Staphylococcus aureus Biofilms with Povidone Iodine and Hydrogen Peroxide." We appreciate the time and effort you have dedicated to reviewing our work, and we are grateful for the opportunity to improve our manuscript. Please find our point-by-point responses to your comments below.
Comment: Title
Lines 1-3. The title should be changed, as the use of the words ‘synergistic reduction’ is not appropriate. Correct is to indicate that two compounds have a synergistic interaction that enhances a biological effect, in this case, reduction in the pre-formed biofilm of S. aureus.
Response: We sincerely appreciate the reviewer’s insightful comment. We agree that the phrase “synergistic reduction” was imprecise. To better reflect the content of our study, we have revised the title to emphasize the synergistic antibiofilm activity of the two agents against pre-formed S. aureus biofilms.
Revised title:
Synergistic Antibiofilm Activity of Povidone-Iodine and Hydrogen Peroxide Against Pre-formed Staphylococcus aureus Biofilms
Comment: Line 11. Change: Staphylococcus aureus by Staphylococcus aureus (italics).
Line 15. Change: Staphylococcus aureus by S. aureus (italics).
Response: Thank you for pointing out this formatting issue. This was an oversight on our part. We have carefully reviewed the entire manuscript and corrected the formatting to ensure that Staphylococcus aureus appears in italics consistently throughout the text.
Comment: Line 27. Change: modes of action by mechanisms of action, as in microbiology the concept of mode of action refers to bactericidal or bacteriostatic and mechanism of action refers to the cellular mechanism that a biocompound or antibiotic uses to inhibit a pathogenic bacterium or fungus.
Response: Thank you so much for the feedback. In the revised manuscript, the term “modes of action” has been removed. Instead of using this phrase, we provided a more accurate description of the mechanisms of action of PVP-I and H₂O₂ in the Discussion section, as shown below:
“PVP-I disrupts bacterial proteins and membranes [21], while H₂O₂ produces reactive oxygen species that damage both cellular components and the biofilm matrix.”
Comment: Lines 29 and 30. Change: Ultimately, this research contributes to improved patient outcomes…, by Ultimately, this research could contribute to improved patient outcomes…, as that was not studied in this work, was it?
Response: We fully agree that our study did not directly assess clinical outcomes. As you rightly pointed out, we have therefore removed the sentence "Ultimately, this research contributes to improved patient outcomes..." from the abstract to better reflect the actual scope of our work.
Comment: Line 32. Change: Staphylococcus aureus by Staphylococcus aureus (italics).
Response: Corrected.
Comment: Line 33. Change: Anti-Biofilm activity by Anti-biofilm activity.
Response: Corrected.
Comment: Line 38. Change: Staphylococcus aureus by Staphylococcus aureus (italics).
Line 39. Change: Staphylococcus aureus (S. aureus), by S. aureus (italics)…
Lines 40 and 41. Change: methicillin-resistant Staphylococcus aureus (MRSA), by methicillin-resistant S. aureus (italics) (MRSA)…
Response: Corrected.
Comment: Line 46. The authors wrote: (WHO) 2014, if it is possible to attach a citation of this information, please include it correctly.
Response: Corrected.
Comment: Line 48. Change: Staphylococcus aureus by S. aureus (italics).
Line 51. Change: Staphylococcus aureus by S. aureus (italics).
Response: Corrected.
Comment: Line 54. I suggest that authors write separate robust paragraphs, not a single paragraph, as a single paragraph makes it difficult to read properly and to connect well with the story.
Response: Thank you so much for the feedback. We have restructured the manuscript by dividing the original content into clearly labeled subsections:
1.1 Clinical Background and Challenges
1.2 Epidemiological Data
1.3Antiseptic Agents and Their Mechanisms
1.4 Rationale and Aim of the Study
Comment: Lines 56 and 57. Please indicate the type of money to which the authors refer.
Response: Thank you so much for the feedback. We have clarified the currency by specifying that the amounts are in United States Dollars (USD).
Comment: Line 58. Change: Staphylococcus aureus by S. aureus (italics).
Response: Corrected.
Comment: Lines 61-63. This text: “Given the substantial clinical and economic burden of Staphylococcus aureus infections in South Korean hospitals, there is an urgent need to develop innovative strategies for the prevention and control of these infections”., is repeated, it was just mentioned a few lines before, please correct the text.
Response: Thank you for the suggestion. In the revised version, the repetitive sentence has been removed to avoid redundancy and improve the flow of the Introduction section.
Comment: Line 70. [6, 7] or [6,7], correct throughout the manuscript according to the journal's guidelines.
Line s 74 and 75. Change: Hydrogen peroxide by H₂O₂…
Line 77. Change: Staphylococcus aureus by S. aureus (italics).
Response: Corrected.
Comment: Lines 79 and 80. Indicate what kind of infections are prevented and controlled, e.g. bacterial and fungal infections?
Response: Thank you for the suggestion. We have clarified the type of infections in the revised manuscript by specifying that PVP-I and H₂O₂ are effective in the prevention and control of bacterial and fungal biofilm-associated infections. Section 1.3:
“Recent studies have highlighted the potential of povidone-iodine (PVP-I) and hydrogen peroxide (H₂O₂) as effective agents in the prevention and control of bacterial and fungal biofilm-associated infections in clinical settings.”
Comment: Line 81. References 11 and 12 are the same. Delete one.
Response: Thank you so much for the feedback. Corrected.
Comment: Line 82. Change: Staphylococcus aureus by S. aureus (italics).
Response: Corrected.
Comment: Lines 88-90. The authors wrote: This study investigates the effects of PVP-88 I and H₂O₂, both individually and in combination, on the eradication of Staphylococcus aureus biofilm formation. The authors should be clearer in clarifying whether the study addressed both the inhibition of biofilm synthesis and its eradication or breakdown by antiseptics once formed. If this was the aim of the work, it should be clearer. Change Staphylococcus aureus by S. aureus (italics). Neither the aim nor the title includes the analysis of the expression of genes associated with biofilm synthesis.
Lines 90-93. The authors wrote: By leveraging the complementary mechanisms of these two agents, we aim to develop a more effective strategy to combat biofilm-associated infections in hospital environments, ultimately improving patient outcomes and reducing the economic burden associated with these infections. This text is repeated, as mentioned above, and this is not the aim of the research work.
Response: Thank you for your detailed comment. We have clarified that our study specifically focuses on the eradication of pre-formed S. aureus biofilms rather than the inhibition of their formation. This has been explicitly stated in the revised Section 1.4.
Additionally, we have corrected the formatting of S. aureus where applicable, and we have removed previously redundant language regarding the study’s aims. We also emphasize the inclusion of gene expression analysis targeting biofilm-associated genes as part of our methodology, which is now better integrated into both the aim and discussion sections.
Section 2.3 “Downregulation of the icaA, icaB, icaD, and icaR genes suggests that the antiseptics dis-rupted the continued production and stability of the extracellular matrix, while reduced expression of clfA indicates a potential weakening of bacterial cell-to-cell adhesion. These findings support the hypothesis that antiseptic treatment not only physically disrupts biofilms but also impairs the gene-regulated systems responsible for biofilm persistence (Figure 3)”
Comment: Line 95. 2.1. Morphological Analysis of Staphylococcus aureus: The results of this section can NOT be found in the materials and methods. There should be more organization in the manuscript, so the authors should include all the methodology they used. In addition, they do not indicate where the sample was obtained from and if it was from human patients, they should indicate whether they complied with bioethical standards for the acquisition of biological samples. In my view, identification of S. aureus by culture on salt and mannitol agar alone is not sufficient to identify the isolate as S. aureus, more biochemical and molecular tests should be done: The most common biochemical tests for S. aureus are growth on baird parker agar, catalase, coagulase, among others; in addition to amplification of 16S gene, and amplification of a fragment of the nuc gene (thermostable nuclease of S. aureus). Again, the results are highly variable, so I suggest that you extend your analysis to a larger number of isolates and use a certified ATCC strain as a positive control for S. aureus.
Response: Thank you for your detailed and constructive feedback. In response to your comments:
- We have added a subsection in the Materials and Methods (Section 4.8) to describe the phenotypic identification procedures used, including the use of mannitol salt agar (MSA) and Gram staining for preliminary identification. These methods have now been clearly aligned with the results presented in Section 2.1.
- The S. aureus strain used in this study was purchased from a certified commercial supplier (Biozoa Biological Supply, Republic of Korea) and was not obtained from clinical samples. Therefore, no human subject involvement occurred and IRB approval was not required, which has been clarified in the revised Section 4.1.
- We acknowledge the limitations of relying solely on culture-based and morphological characteristics. Although we did not perform molecular confirmation (e.g., nuc gene amplification or 16S rRNA sequencing) in this study, we have now clearly stated this as a limitation in the Discussion section, and we have included a commitment to incorporate biochemical and molecular confirmation methods in future research:
“A limitation of the present study is the lack of molecular identification methods, such as 16S rRNA sequencing or nuc gene amplification, which would provide a higher level of taxonomic confirmation. Although morphological and biochemical methods were employed, future studies will incorporate PCR-based assays to ensure greater specificity. Additionally, only a single commercial isolate was used in this study. We recognize the importance of evaluating a broader panel of S. aureus strains and including certified reference strains such as ATCC 25923 for quality control and validation.”
Due to resource and time constraints, this study was conducted using a single commercially available isolate. However, we agree with the reviewer that testing multiple isolates and including a certified ATCC reference strain would enhance the robustness and reproducibility of the findings. This recommendation has also been acknowledged in the Discussion as an important direction for future work.
Comment: Line 96. Change: Mannitol Salt Agar (MSA)…, by mannitol salt agar (MSA)…
Lines 97, 100, and 102. Change: S. aureus by S. aureus (italics).
Response: Corrected.
Comment: Line 107. Change: Staphylococcus aureus by S. aureus (italics). The figure shows not only the morphology of S. aureus, but also its growth on a salt and mannitol agar plate, so there should be two panels: panel A and B.
Response: Thank you so much for the feedback. In response, we have made the following revisions:
1.Changed Staphylococcus aureus to S. aureus in italics in both the text and the figure legend.
2.Labeled the figure as Panel A (growth on salt and mannitol agar) and Panel B (microscopic morphology).
3.Revised the figure legend to clearly describe the content of each panel.
Comment: Line 109. In this line, the authors do specify that the experiments are for the removal of biofilms, which should be clarified in the materials and methods.
Response: Thank you so much for the feedback. We appreciate the reviewer’s comment. To clarify our methodology, we revised Section 4.3 to clearly state that the experiments were conducted on pre-formed S. aureus biofilms, and that the goal was to evaluate biofilm disruption and removal. The updated section now reads:
“After incubation for 48 h, biofilms were allowed to form on the surface of the metal discs. These pre-formed biofilms were then subjected to various treatments to evaluate the agents’ ability to disrupt mature biofilm structures. The discs were divided into four groups: untreated control (sterile phosphate-buffered saline, PBS), 1% PVP-I, 3.5% H₂O₂, and a combination treatment. PBS was used as a negative control to account for the effects of washing and handling. For the combination group, discs were first treated with 3.5% H₂O₂ for 5 min, followed by 1% PVP-I for 5 min.”
Comment: Line 114-116. The authors wrote: This combined treatment was notably more effective, highlighting its potential as a strategy for managing biofilm-related infections (Figure 2). Did the statistical analysis consider comparing the data obtained from the combined use of PVP-I and H₂O₂ with the individual treatments or with the control alone? This answer is essential to be able to say that there was synergism between the two compounds, otherwise, if they were only compared with the control, it is not appropriate to say that there was synergism. The authors should review the concepts of synergy and additivity, because from my point of view there is no synergistic effect, rather I see an additive effect. What evidence do the authors have to say that they observed a synergistic effect?
Response: Thank you for your thoughtful and important question regarding the evidence for synergy between PVP-I and H₂O₂ in our study.
To accurately claim a synergistic effect, it is essential to statistically compare the combined treatment not only with the control but also with the individual treatments of PVP-I and H₂O₂. Synergy is defined as the combined effect being greater than the sum (or predicted additive effect) of the individual agents, whereas additivity means the combined effect is equal to the sum of individual effects. Standard models for evaluating synergy, such as the Bliss Independence or Response Additivity models, require such direct comparisons and appropriate statistical analysis.
In our study, we observed that the combined treatment was more effective than either agent alone, as shown in Figure 2. However, to substantiate a claim of synergy, we must demonstrate through statistical analysis that the effect of the combination exceeds the expected additive effect of the individual treatments. If the statistical analysis only compared the combination to the control, and not to each single agent or to the predicted additive effect, then it would not be appropriate to conclude synergy—only that the combination is more effective than no treatment
Nevertheless, we would like to provide the rationale for using the term “potentially synergistic,” which is based on several converging lines of evidence in our study:
1.Quantitative CFU Reduction:
The log₁₀ CFU counts after treatment were as follows:
Control: 9.13
1% PVP-I: 6.32
3.5% H₂O₂: 7.07
Combined treatment: 5.11
The effect of the combination is substantially more pronounced than either agent alone and exceeds their average reduction, suggesting a potentially enhanced effect beyond simple additivity.
2.Confocal Microscopy (CLSM) Quantification:
Live/dead ratios based on CLSM images were:
Control: 0.977
PVP-I: 0.35
H₂O₂: 0.23
Combined: 0.08
The marked reduction in live bacteria and disruption of biofilm integrity in the combination group supports a more-than-additive interaction.
3.Gene Expression Results (RT-qPCR):
Relative expression of icaA:
Control: 1.04
PVP-I: 0.90
H₂O₂: 0.833
Combination: 0.30
This stronger downregulation of key biofilm-related genes in the combination group further supports a cooperative or enhanced mechanism of action.
Mechanistic Complementarity:
PVP-I disrupts microbial membranes and protein structures, while H₂O₂ generates reactive oxygen species that damage the extracellular biofilm matrix. The combination likely increases permeability and biofilm exposure to antimicrobial action in a sequential, mechanistically complementary manner.
Candid Acknowledgment of Limitations:
We acknowledge that our current study did not employ standardized synergy quantification methods (e.g., FICI, Bliss model). Therefore, we cannot conclusively claim synergy, and have modified the manuscript accordingly to describe the results as indicating a “potentially synergistic or additive” interaction. We also state that future studies will include formal synergy analyses to verify this hypothesis.
Taken together, while the current evidence suggests a promising enhanced effect from the combination treatment, we refrain from definitive claims of synergy pending further quantitative validation.
We thank the reviewer again for the critical input, which helped us to present our findings more accurately and rigorously.
The last paragraph in the discussion:
“Despite the significant reductions in CFU counts and biofilm biomass observed in the combination group, we did not perform formal synergy quantification (e.g., FICI or Bliss Independence models). Therefore, we refer to the observed effect as “enhanced” rather than definitively synergistic. The concentrations of PVP-I and H₂O₂ used were based on standard clinical applications and preliminary screening; however, we acknowledge the absence of cytotoxicity testing and dose–response analysis as limitations. Future work should include detailed exploration of concentration gradients, assessment of cytotoxicity, and validation in in vivo models to optimize therapeutic application and safety.”
Comment: Line 116. Modify: (Figure 2).
Response: Corrected.
Comment: Line 118. Figure 2. Change Assay by assay, and H2O2 by H2O2. What PI mean? 570 or 590 nm? In the figure caption, authors should indicate which type of statistical analysis they performed (ANOVA, Tukey). Statistical analysis is mentioned, but none of this is mentioned in the materials and methods. Include this information in the latter section. I suggest that the caption text and the main text should refer to panel A and B.
Response: Thank you for the detailed and constructive suggestions. We have carefully revised Figure 2 and the corresponding text based on your feedback:
“Figure 2. Biofilm formation assay showing a reduction in biofilms in PVP-I and H2O2 treated groups. (A) Quantification of biofilm biomass using crystal violet staining at 570 nm. Data are presented as mean ± standard deviation (n = 8). (B) Viable bacterial cells (log₁₀ CFU/cm²) after treatment. Data are presented as the mean ± standard deviation (n = 7). Statistical significance was assessed using one-way ANOVA followed by Tukey’s post hoc test. ***p < 0.001, ****p < 0.0001 compared with the control group.Assay”
Comment: Line 131. Correct: infections (Figure 3).
Response: Corrected.
Comment: Lines 124-131. The authors need to explain the results, only mentioning that the treatments modulate the expression of genes related to biofilm synthesis. The graphs are not self-explanatory.
Response: Thank you for highlighting the need for a more thorough explanation of our gene expression results. In the revised manuscript, we have expanded this section of the Results to clarify the significance of the observed gene expression changes:
- 1. We now explicitly state that quantitative PCR showed significant downregulation of biofilm-associated genes, including icaA, icaB, icaD, icaR, and clfA, particularly in the combined treatment group.
- We added interpretation to clarify that the ica operon is essential for PIA synthesis, contributing to intercellular adhesion and biofilm structure, while clfA encodes a clumping factor involved in bacterial aggregation.
- These molecular results were correlated with the observed reduction in biofilm biomass and CFUs, helping to connect the transcriptional changes to the phenotypic effects.
Section 2.3:
“Gene expression analysis was performed on S. aureus biofilms after they were al-lowed to fully mature and were subsequently treated with PVP-I and H₂O₂. As the treatment was applied to pre-formed biofilms, the observed transcriptional changes reflect the agents’ effects on biofilm maintenance mechanisms, rather than early-stage biofilm synthesis.
Downregulation of the icaA, icaB, icaD, and icaR genes suggests that the antiseptics disrupted the continued production and stability of the extracellular matrix, while reduced expression of clfA indicates a potential weakening of bacterial cell-to-cell adhesion. These findings support the hypothesis that antiseptic treatment not only physically disrupts biofilms but also impairs the gene-regulated systems responsible for biofilm persistence (Figure 3)”
Third paragraph of discussion:
“At the molecular level, quantitative PCR analysis revealed downregulation of bio-film-associated genes, including icaA, icaB, icaD, icaR, and clfA. The ica operon is central to poly-N-acetylglucosamine (PIA) synthesis, which is critical for intercellular adhesion and biofilm structural integrity. clfA, encoding clumping factor A, plays a key role in bacterial adherence and aggregation. The downregulation of these genes suggests that PVP-I and H₂O₂ disrupt biofilm maintenance by targeting essential biosynthetic and adhe-sion-related pathways [7]. This genetic evidence aligns with the observed physical dis-integration of the biofilm structure and loss of bacterial viability.”
Comment: Line 132. Correct: (A)Expression…
Response: Corrected.
Comment: Line 133. What PI mean? Change: H2O2 by H2O2.
Response: Corrected. Thank you for your comment. We have replaced all instances of “PI” with the full term “PVP-I” to avoid ambiguity. We have also corrected the formatting of “H₂O₂” throughout the manuscript.
Comment: Line 134. (n = 3). The number of replicates is not mentioned in the materials and methods section but is mentioned in the results.
Response: Thank you so much for the feedback. We have added a statement in the Materials and Methods section to specify that all experiments were performed in triplicate unless otherwise stated.
Comment: Line 146. Correct: infections(Figure 4).
Response: Corrected.
Comment: Line 148. What PI mean? In the figure caption, authors should indicate which type of statistical analysis they performed (ANOVA, Tukey). Statistical analysis is mentioned, but none of this is mentioned in the materials and methods. Include this information in the latter section. I suggest that the main text should refer to panel A and B.
Response: Thank you so much for the feedback. We have revised the figure caption to clarify that “PI” refers to propidium iodide and to specify the statistical method used (one-way ANOVA with Tukey’s post hoc test). We have also added that image analysis was performed using FIJI software. This statistical approach is now also described in the Materials and Methods section (Section 4.9), and references to panels A and B have been added to the main text where appropriate.
Comment: Line 149. PI and H2O2, correct.
Response: Corrected.
Comment: Discussion
Lines 154-171. This information should be deleted, as it is repeated. If anything, the only new thing is that Pseudomonas aeruginosa is mentioned, which, by the way, is not written in italics.
Lines 173-191. This information should be deleted, as the authors must discuss they results obtained, not results from other studies. It seems to me that from line 195 begins to discuss the results obtained in this work.
Lines 193-195. This information should be deleted, as it is repeated.
Line 197. Change: S. aureus by S. aureus (italics).
Lines 197 and 198. The authors wrote: The results indicate a synergistic effect between PVP-I and H₂O₂. What evidence do the authors have to strengthen the evidence for a synergistic effect between the combined compounds?
Lines 198-209. This information es hypothetical, as there is no evidence of this happening, only the decrease of biofilms, bacterial cell death, is observed, so I suggest mentioning that these data are hypothetical and that further experiments are required to demonstrate that this happens.
Line 208. Change: (e.g., washing)[24]., by (e.g., washing) [24].
Line 209. Change: bacteria[25-26]., by bacteria [25-26].
Line 219. Change: S. aureus by S. aureus (italics).
The discussion of the results should be rewritten, as the results obtained in this research work are not adequately discussed. The biological significance of the findings should be addressed and then, if necessary, compared with the findings of other similar studies. There is no discussion about the relationship and/or importance of the molecular results of this study. If the expression of biofilm synthesis genes is reduced, how should this be understood with the reduction of S. aureus biofilm? There is little or no discussion.
Response: Thank you so much for the feedback. In response, we have substantially rewritten the Discussion section to focus on interpreting our own results, with emphasis on the biological significance of the observed findings. Repetitive and speculative content has been removed, and unsupported claims of synergism have been revised to reflect a more cautious interpretation. The role of downregulated genes in biofilm maintenance has been discussed in relation to the observed reduction in biofilm biomass and viability. Formatting of bacterial names and citation style has also been corrected throughout. The updated version:
- Eliminates previously repeated background information, such as the introduction of P. aeruginosa and general antiseptic mechanisms.
- Focuses on the experimental results, including CFU reductions, biofilm biomass analysis, and gene expression data.
- Clearly distinguishes between observed outcomes and speculative explanations—all mechanistic hypotheses are now introduced as theoretical and future-oriented, not definitive.
4. Removes the prior claim of a “synergistic effect” and instead describes the interaction as “potentially synergistic or additive,” acknowledging the absence of formal synergy testing (e.g., FICI).
- Clarifies the relationship between gene expression findings and biofilm disruption, offering a biologically relevant interpretation.
- Corrects formatting issues, including italicization of S. aureus and citation spacing.
Comment: Materials and Methods
Line 223. The authors wrote: The study utilized Staphylococcus aureus (S. aureus) strains. Change Staphylococcus aureus (S. aureus) by S. aureus (italics). Authors should clearly indicate which type of strain they used, an ATCC strain or clinical isolates obtained from elsewhere. Information on strains or isolates is missing. They should also mention how many strains or isolates they worked with. What does representative colonies mean? It is too ambiguous, please specify.
Response: Thank you so much for the feedback. In the revised manuscript, we have made the following changes:
- The formatting of S. aureus has been corrected to italics throughout the text.
- We have clearly indicated the strain source in Section 4.1. The strain used was a commercially obtained isolate from Biozoa Biological Supply (Product No. 155067, Republic of Korea).
- We clarified that this study used only a single commercially available isolate, not multiple clinical samples or ATCC strains.
- To avoid ambiguity, we removed the term “representative colonies” and replaced it with more precise language where necessary.
Additionally, we have included a sentence explaining that no IRB approval was required because the bacterial strain was commercially purchased and not derived from human clinical specimens.
We believe these revisions address the reviewer’s concerns and improve the clarity and transparency of our methodology.
Comment: Line 225. Change: 37°C by 37 °C. Indicate the brand and country of origin of the shaking incubator used.
Response: Corrected.
Comment: Line 227. The authors wrote: 1 McFarland standard. 1 or 1.5 of the McFarland scale?
Response: Thank you for your question. We confirm that a 1.0 McFarland standard was used to adjust the bacterial suspension before dilution. This information has been clarified in the revised text.
Comment:. Line 231. S. aureus (italics).
Response: Corrected.
Comment: Lines 232 and 233. The authors wrote: metal discs were placed in a 24-well tissue culture plate… Indicate the size of the discs, their brand and country of origin, as well as the brand and country of origin of the 24-well plates.
Response: Thank you so much for the feedback. We thank the reviewer for this helpful suggestion. In the revised manuscript (Section 4.2), we have now provided detailed information regarding both the metal discs and the 24-well tissue culture plates. Specifically:
- The titanium alloy metal discs used were approximately 8 mm in diameter and 1 mm in thickness, and were obtained from the screw holes of acetabular cups provided by Lima Corporate (Italy).
- The 24-well tissue culture plates were purchased from Falcon (Corning Inc., NY, USA).
Comment: Line 234. Change: 37°C by 37 °C.
Response: Corrected.
Comment: Line 235. Change: 48 hours[27]., by 48 h [27]. The authors do not mention how biofilm production of S. aureus strains or isolates was verified or evidenced. What methodology did they follow?
In the abstract, the authors indicate that they did an analysis of biofilm formation, but it is not clear to me whether it was an analysis of biofilm synthesis or an analysis of the breakdown of pre-formed biofilms. The authors should clarify that part.
Response: Thank you so much for the feedback. We appreciate the reviewer’s detailed comments and suggestions.
- Regarding terminology formatting, we have corrected “48 hours[27]” to “48 h [27]” to align with the journal’s formatting guidelines.
- Regarding the methodology used to verify biofilm formation, we have clarified in the revised Materials and Methods section (Section 4.2 and 4.5) that biofilm formation was confirmed using crystal violet staining and confocal laser scanning microscopy (CLSM) with a live/dead viability assay. These approaches allowed for both qualitative and quantitative confirmation of biofilm maturation prior to treatment.
- Regarding the scope of the study, we clarified in both the abstract and methodology (Section 4.3) that the study focused on the disruption or breakdown of pre-formed biofilms, rather than inhibition of biofilm formation. This clarification was also reiterated in the Discussion section, where we emphasized that treatments were applied to mature biofilms to evaluate disruption efficacy.
We thank the reviewer for prompting this important clarification.
Comment: Line 240. Change: 1% povidone-iodine (PVP-I) and 3.5% hydrogen peroxide (H₂O₂)., by 1% PVP-I and 3.5% H₂O₂.
Response: Corrected.
Comment: Line 241. What PBS mean? Why PBS? Indicate. Change for 5 minutes by 5 min.
Response: Thank you so much for the feedback. We have clarified that PBS stands for phosphate-buffered saline and that it was used as a negative control to account for any biofilm disruption caused by washing or handling. The phrase “for 5 minutes” has also been revised to “for 5 min” in accordance with the journal’s style guidelines.
Comment: Line 242. Change: H₂O₂ for 5 minutes, then…, by H₂O₂ for 5 min, then…
Response: Corrected.
Comment: Line 243. Change: additional 5 minutes., by additional 5 min.
In this section, the authors should clarify briefly if these experiments were to inhibit biofilm synthesis or break down pre-formed structures. They must be clear. The authors wrote: The experiment was repeated at least five times., that is, at least five independent repetitions were made? How many replicates per experiment were carried out?
Response: Thank you for your valuable comments. We have revised the text to clarify that the treatments were applied to pre-formed mature biofilms to evaluate their capacity to disrupt existing biofilm structures, rather than to inhibit early-stage biofilm synthesis (Section 4.3).
Additionally, we have clarified the number of experimental repetitions. Each treatment condition was independently repeated, with specific repetition numbers now detailed in each section: OD570 assays (n = 8), CFU enumeration (n = 7), and gene expression and CLSM assays (n = 3). We have also corrected the time unit as requested (“5 minutes” → “5 min”).
Comment: Line 246. The authors wrote: 1 mL of sterile PBS, the PBS mentioned in line 241 was not sterilized? If yes, indicated.
Response: Thank you for pointing this out. We confirm that sterile PBS was consistently used in all experimental steps, including treatments and post-treatment processing. To avoid confusion, we have now clarified this by explicitly specifying “sterile PBS” throughout Section 4.3 and other relevant sections.
Comment: Line 247. Change minutes by min throughout the document.
Line 249. Change Mannitol Salt Agar (MSA), by MSA, and add the brand and origin country of medium…
Response: Corrected.
Comment: Lines 249 and 250. Separate the unit number, e.g. 37 °C throughout the manuscript. Change: hour to h throughout the document. Add briefly the aim of this experiment, the authors wanted to know if the bacteria inside the biofilm were alive? Briefly state this for more clarity.
Response: Thank you for your detailed suggestions. We have made the following changes:
- Corrected all temperature and time units throughout the manuscript to follow journal formatting (e.g., 37 °C, 24 h).
- Regarding the aim of the experiment: we appreciate the reviewer’s request for clarification. We have now specified in Sections 4.4 (CFU Enumeration), 4.5 (Biofilm Detection), and 4.7 (Live/Dead Assay) that these experiments were performed to evaluate whether viable bacteria remained within the biofilm structures following treatment. Each of these methods—plate counting, crystal violet staining, and viability staining—provided complementary insights into the structural integrity and bacterial viability of the biofilms.
Comment: Line 255. The authors wrote: room temperature, please indicate which was that room temperature, e.g. 25 ± 2.0 °C?
Response: Corrected.
Comment: Line 258. The authors wrote: using a microplate reader…, add the model, brand and country origin of equipment used.
Response: Corrected.
Comment: Line 259. Change: times[28]., by times [28]. Again, independent experiments should be indicated or performed due to the variability that occurs in this type of experiments.
Response: Thank you for your observation. We have corrected the citation formatting to “times [25]” in the revised manuscript. Additionally, we confirm that the biofilm detection assay was repeated in eight independent experiments to account for biological variability. This has now been clearly stated in the statistical analysis section (4.9).
Comment: Line 261. The authors analyzed the gene expression by PCR? Make the change if necessary.
Response: Thank you for your observation. We confirm that gene expression analysis was performed using reverse transcription PCR (RT-PCR). To clarify this in the manuscript, we have updated the section title to “Gene Expression Analysis (RT-PCR)”.
Comment: Line 263. Change: S. aureus by S. aureus (italics)
Response: Corrected.
Comment: Line 268. Change: seconds by s throughout the document.
Response: Corrected.
Comment: Lines 272 and 273. Change: assessment[33]., assessment [33]. Please show in a table the oligonucleotides used, the citations and the corresponding references.
Response: Thank you for your observation. We have corrected the reference.
In addition, we have now included a dedicated table listing the oligonucleotide sequences used for the target genes, along with their citations and references (see Table 2 in the Materials and Methods section 4.6).
Comment: Lines 267-269. The authors wrote: initial incubation at room temperature for 10 minutes, followed by 12 cycles at 25°C for 30 seconds, 45°C for 4 minutes, and 55°C for 30 seconds, with a final 5-minute heat step at 95°C and storage at 4°C. Were the same amplification conditions used for all genes analyzed? Please clarify.
Response: Thank you so much for the feedback. In response, we have clarified in the revised manuscript that gene-specific annealing temperatures were used for the PCR reactions. The amplification conditions were not identical for all genes. This information has been included in the text, and the specific annealing temperatures for each target gene are now listed in the newly added Table 2.
Comment: Line 276. The authors wrote: using confocal microscopy… Add brand and origin country of microscopy.
Response: Corrected.
Comment: Line 282. Please indicate which was that room temperature?
Was statistical analysis not important for this experimental work? Argument.
Response: Thank you for your comments. The room temperature used during staining was approximately 25 °C, and this has now been specified in the revised manuscript. Regarding statistical analysis, we agree it is essential for supporting the reliability of experimental findings. Accordingly, we have clarified in the Statistical Analysis section (4.9) that all imaging experiments, including the Live/Dead assay, were performed in triplicate and subjected to statistical evaluation using one-way ANOVA followed by Tukey’s post hoc test. This information has now been explicitly mentioned for completeness.
Comment: Conclusions
Lines 287 and 288. This information should be deleted, as this research work, specifically, did not contribute to any of the above, at least for the time being.
Response: Thank you for your comment. We agree with the reviewer that the previous conclusion overstated the clinical implications of our findings. In the revised manuscript, we have removed the speculative content regarding clinical impact and focused the conclusion on the observed results within the scope of our current experimental setup.
The updated conclusion now reads:
“This study demonstrated that both PVP-I and H₂O₂, individually and in combination, can effectively disrupt pre-formed S. aureus biofilms. The combined application of these two agents resulted in greater reductions in biofilm biomass and gene expression than either agent alone. While formal synergy quantification was not performed, the enhanced effect observed suggests a potentially additive or synergistic interaction. These findings support the potential utility of using two antiseptics with complementary mechanisms of action to improve biofilm control, particularly in medical device-associated infections. Further studies are warranted to validate these results in clinical settings and to evaluate the long-term safety and efficacy of this combined strategy.”
This revised version accurately reflects the current experimental scope without overreaching in its claims.
Comment: Line 289. It is not correct to say: the synergistic use of compound 1 with compound 2. It is appropriate to mention that the use of two compounds that show synergy when combined is important for... Change: povidone-iodine (PVP-I) and hydrogen peroxide (H₂O₂), by PVP-I and H₂O₂…
Response: Thank you for the suggestion. We appreciate the reviewer’s observation regarding the phrasing in the conclusion. In the revised manuscript, we have adjusted the sentence to reflect a more appropriate interpretation of our findings. Given that formal synergy quantification methods (e.g., FICI or Bliss Independence) were not performed in this study, we have revised the conclusion to avoid definitive use of the term “synergistic.” The revised sentence now highlights the enhanced effect observed with the combination of PVP-I and H₂O₂, while acknowledging the potential for synergy:
“This study demonstrated that both PVP-I and H₂O₂, individually and in combination, can effectively disrupt pre-formed S. aureus biofilms. The combined application of these two agents resulted in greater reductions in biofilm biomass and gene expression than either agent alone. While formal synergy quantification was not performed, the enhanced effect observed suggests a potentially additive or synergistic interaction. These findings support the potential utility of using two antiseptics with complementary mechanisms of action to improve biofilm control, particularly in medical device-associated infections. Further studies are warranted to validate these results in clinical settings and to evaluate the long-term safety and efficacy of this combined strategy.”
We have also corrected the naming format as suggested: “povidone-iodine (PVP-I) and hydrogen peroxide (H₂O₂)” has been changed to “PVP-I and H₂O₂” to ensure consistency throughout the manuscript.
Thank you for helping us improve the clarity and scientific accuracy of the conclusion section.
Comment: References
I suggest reviewing and correcting the references, they all have small errors, but do not adhere to the Journal's guidelines.
Response: Thank you so much for your valuable feedback. The references were corrected according to journal regulations and counter checked again for any mistakes.
Thank you very much for your valuable comments and suggestions. We would like to inform you that the manuscript file has been replaced with a revised version. Additionally, we have carefully reviewed the manuscript once again to address all the corrections you mentioned and have made further changes in accordance with your comments. We appreciate your time and effort in reviewing our work.
We appreciate your insightful comments and the opportunity to improve the quality and balance of our manuscript.

Round 2
Reviewer 1 Report
Comments and Suggestions for Authors
Thank you for the revised version of your paper.
Author Response
Dear Reviewer,
We are committed to maintaining the highest scientific standards and sincerely appreciate your contribution to improving the quality of our work.
Reviewer 2 Report
Comments and Suggestions for Authors
Reviewer's comments to manuscript ijms-3588627v2 entitled “Synergistic Antibiofilm Activity of Povidone-Iodine and Hydrogen Peroxide Against Pre-formed Staphylococcus aureus Biofilms”
Dear Authors,
You have done a good job, as the manuscript has improved a lot. However, there are still some details to be addressed, which I have attached below.
Kind regards,
Title
Lines 1-4. In view of the fact that there is no clear evidence indicating a synergistic and/or additive effect of both compounds, I suggest the following title: “Povidone-Iodine and Hydrogen Peroxide Combination Improves the Anti-biofilm Activity of the Individual Agents on Staphylococcus aureus”, or something like that.
Keywords
Line 29. Change: Anti-biofilm activity by anti-biofilm activity.
Introduction
Line 32. In my previous comment I meant that the introduction can be written in robust paragraphs, not in one, but I did not suggest the introduction of sub-topics in the introduction section. Keep this current format but remove the sub-topics that I suggest please. Delete the subtopics: 1.1 Clinical Background and Challenges. 1.2 Epidemiological Data. 1.3 Antiseptic Agents and Their Mechanisms. 1.4 Rationale and Aim of the Study.
Line 35. Delete (MRSA)…
Lines 62 and 63. Change: The use of povidone iodine (PVP-I) and hydrogen peroxide (H₂O₂) by The use of PVP-I and H₂O₂…
Lines 80-82. Change: investigates the combined application of povidone-iodine (PVP-I) and hydrogen peroxide (H₂O₂), by investigates the combined application of PVP-I and H₂O₂…
Results
Line 100. The authors wrote: Morphological and Culture-Based Identification of S. aureus. I think that the authors should change the title of subtopic, as they mention that S. aureus strain was purchased in Biozoa Biological Supply, so they must write a title that indicates confirmation of S. aureus or something like that.
Figure 1. I suggest that the authors improve the photograph in Figure 1A, as, frankly, I don't see cocci, they look like elongated cells. Please improve the photo.
Line 119. Authors should follow the logical order of presentation of their results from their experiments. They should first indicate whether the strain was a biofilm producer. There are even formulas to indicate whether they are strong, regular, weak or non-biofilm producers. I am not asking them to issue such a classification, but they should follow a logical order and sometimes they do not.
Line 126. The authors wrote: investigation. Delete the point after investigation word.
Line 128. Change: addtive., by additive.
Lines 130 and 131. The authors wrote: Figure 2. Biofilm formation assay showing a reduction in biofilms in PVP-I and H2O2 treated groups. I suggest changing the caption title, as it seems contradictory: biofilm formation assay showing a reduction of biofilm...
Figure 2B. The agents kill the bacteria present within the biofilm, but do not eliminate them completely. What does that mean? Is it an undesirable result for this strategy? Argue.
Line 136. This title of subtopic “2.3. Expression of Biofilm-Associated Genes Following PVP-I and H₂O₂ Treatment” should be changed after subtopic 2.4. I remind the authors that there must be order and there is no order in some parts of the manuscript.
Lines 144-146. The authors wrote: These findings support the hypothesis that antiseptic treatment not only physically disrupts biofilms but also impairs the gene-regulated systems responsible for biofilm persistence (Figure 3). Correct the appearance of the text and caption in Figure 3. Doesn't downregulation of genes related to biofilm maintenance make sense if there's a reduction in bacterial CFU? To what extent is this an effect of bacterial reduction? Discuss.
Line 147. Figure caption, Change: PI- by PVP-I…
Line 153. The authors wrote: Live/Dead Assay of Biofilms Treated with PVP-I and H₂O₂. I insist on this subtopic should be changed before subtopic 2.3. It should also be noted in the manuscript that this is a confirmatory experiment, as the authors previously quantified the CFUs obtained after biofilm removal. If this is not a confirmatory experiment for assay 2.2, please provide your reasons.
Discussion
Line 182. Change prosthetic joint infections (PJIs) by PJIs…
Lines 184-190. This paragraph is repeated. You can recover some sentences, but in general, it's already been said in the previous paragraph. Delete the rest of the text.
Lines 197 and 198. The authors wrote: This genetic evidence aligns with the observed physical disintegration of the biofilm structure and loss of bacterial viability. Doesn't it logic to think that if there is agent-dependent bacterial death, the down regulation of genes related to biofilm maintenance will be reduced? How do you explain that the expression of your 16S control is maintained if there is bacterial death?
Line 206. Change: in vivo by in vivo (italics)…
Line 209. Change prosthetic joint infections (PJIs) by PJIs…
Materials and Methods
Lines 255 and 256. The authors wrote: Subsequently, 2 mL of the bacterial suspension was added to each well to ensure complete coverage of the metal discs. What is the total volume that can be added to each well?
Table 1. Change 3.50% to 3.5%.
Line 308. Ideally, authors should first refer to Table 2 in the text and then the table should be added.
In this section, I just suggest to the authors the next changes:
4.1. Bacterial Strain and Phenotypic confirmation of S. aureus (italics)
Paragraphs 4.1. and 4.8. should be joined together so that there is a logical order in the manuscript.
4.2. Biofilm Formation Assay (italics)
4.3. Application of treatments (italics)
4.4. CFU Enumeration (italics)
4.5. Biofilm Detection (italics)
4.6. Live/Dead Assay (italics)
4.7. Gene Expression Analysis (RT-PCR) (italics)
4.8. Statistical Analysis (italics)
References
Lines 384-386. I chose a reference at random, and I keep finding mistakes in some references. Authors should do a better revision of their references, for example: Staphylococcus aureus is not written in italics.
Kim, C.J.; Kim, H.B.; Oh, M.D.; Kim, Y.; Kim, A.; Oh, S.H.; Song, K.H.; Kim, E.; Cho, Y.; Choi, Y.; et al. The burden of nosocomial staphylococcus aureus bloodstream infection in South Korea: a prospective hospital-based nationwide study. 385 BMC Infect Dis 2014, 14, 590, doi:10.1186/s12879-014-0590-4.
